# Impact of the COVID-19 pandemic on utilisation of facility-based essential maternal and child health services from March to August 2020 compared with pre-pandemic March–August 2019: a mixed-methods study in North Shewa Zone, Ethiopia

Chalachew Bekele ![ORCID],[1] Delayehu Bekele,[2] Bezawit Mesfin Hunegnaw,[3] Kimiko Van Wickle,[4] Fanos Ashenafi Gebremeskel,[5] Michelle Korte,[6] Christine Tedijanto,[7] Lisanu Taddesse,[8] Grace J Chan ![ORCID] [4,9]

For numbered affiliations see end of article.

**Correspondence to**
Dr Grace J Chan;
grace.chan@hsph.harvard.edu

## ABSTRACT

**Introduction** Health systems are often weakened by public health emergencies that make it harder to access health services. We aimed to assess maternal, newborn and child health (MNCH) service utilisation during the first 6 months of the COVID-19 pandemic compared with prior to the pandemic.

**Methods** We conducted a mixed study design in eight health facilities that are part of the Birhan field site in Amhara, Ethiopia and compared the trend of service utilisation in the first 6 months of COVID-19 with the corresponding time and data points of the preceding year.

**Result** New family planning visits (43.2 to 28.5/month, p=0.014) and sick under 5 child visits (225.0 to 139.8/ month, p=0.007) declined over the first 6 months of the pandemic compared with the same period in the preceding year. Antenatal (208.9 to 181.7/month, p=0.433) and postnatal care (26.6 to 19.8/month, p=0.155) visits, facility delivery rates (90.7 to 84.2/month, p=0.776), and family planning visits (313.3 to 273.4/month, p=0.415) declined, although this did not reach statistical significance. Routine immunisation visits (37.0 to 36.8/month, p=0.982) for children were maintained. Interviews with healthcare providers and clients highlighted several barriers to service utilisation during COVID-19, including fear of disease transmission, economic hardship, and transport service disruptions and restrictions. Enablers of service utilisation included communities' decreased fear of COVID-19 and awareness-raising activities.

**Conclusion** We observed a decline in essential MNCH services particularly in sick children and new family planning visits. To improve the resiliency of fragile health systems, resources are needed to continuously monitor service utilisation and clients' evolving concerns during public health emergencies.

## STRENGTHS AND LIMITATIONS OF THIS STUDY

⇒ We presented data on service utilisation during the early months of the pandemic in a rural, agrarian region in Ethiopia.

⇒ The mixed-methods approach integrated both quantitative service utilisation coverage and exploratory qualitative interviews to understand our findings and the reasons for changes in service utilisation.

⇒ We focused on the coverage of service utilisation as the primary outcome rather than mortality or morbidity rates.

⇒ We do not have detailed data on service provision (eg, which services were restricted and for how long, in what manner).

⇒ Since we collected the qualitative data 3 months past the initial 6 months of the pandemic (March–August 2020), there may be recall bias.

## INTRODUCTION

The WHO declared COVID-19 a global pandemic on 11 March 2020[1] and Ethiopia registered its first case of COVID-19 on 13 March 2020. Ethiopia has reported relatively low numbers of COVID-19 cases and COVID-19-related deaths, with 63 367 confirmed cases and 974 deaths in a population of 119 million, as of 10 September 2020.[2] The majority of reported cases were from the capital city, Addis Ababa, and only 365 confirmed cases and 8 deaths were registered by 30 August 2020 in the North Shewa Zone (the third administration unit of the country), where the study was conducted.

Multiple preventive measures focused on social distancing and wearing masks were undertaken in Ethiopia.[3] Some health facilities were assigned as COVID-19 isolation and quarantine centres, and many health facilities suspended elective surgeries and select outpatient services. This increasing burden of managing COVID-19 on health facilities and healthcare providers (HCPs) left the health system overstretched, challenging its ability to operate effectively. As shown during the 2014–2015 Ebola outbreak in West Africa, when health systems are overwhelmed by outbreaks, mortality from vaccine-preventable and other treatable diseases can increase dramatically.[4 5]

Well-organised and equipped health systems can continue to provide equitable access to essential services through an emergency,[6] but fragile health systems in developing countries face organisational and resource constraints when confronted with emergencies such as pandemics. The WHO advises nations to identify and prioritise maternal, newborn and child health (MNCH) essential services (eg, routine vaccination, reproductive health services, childbirth, and care of young infants and older adults) in their efforts to maintain continuity of service delivery and make strategic shifts to ensure limited resources provide maximum benefit for the population.[7] However, the disruption of services and diversion of resources away from essential sexual and reproductive healthcare due to the prioritisation of the COVID-19 response are expected to increase the risks of maternal and child morbidity and mortality.[8]

A lockdown or stay-at-home policy was largely not in place in Ethiopia where the study was conducted, and health facilities were open during the study period from March to August 2020. The Ministry of Health (MOH) of Ethiopia focused on preventive measures and control of the pandemic. Possible shifts of the health workforce and the health system toward the COVID-19 response may have contributed to low utilisation of routine services. For example, 'half seat' policies (29 March 2020) decreased maximum occupancy on public transit by half and increased the cost of transportation when travelling within the region, while transportation between regions was paused completely. Additionally, school closures (March 2020), the declaration of a state of emergency (08 April 2020), and awareness campaigns about the pandemic and case reports, both suspected and confirmed, may have contributed to the fear of exposure to COVID-19, especially for patients visiting health facilities.

To understand these effects of the COVID-19 pandemic in Ethiopia, the HaSET Maternal and Child Health Research Program assessed trends in MNCH care utilisation from March 2019 to August 2020 as well as HCPs' and clients' perceptions of the barriers to and enablers of service provision and utilisation during the COVID-19 emergency. This study has paramount importance in filling the evidence gap on MNCH service utilisation during COVID-19, both in the Ethiopian context and other low/middle-income countries, to prevent significant damage to the gains achieved in such areas over the past several decades.

## METHODS

We conducted the study in eight health facilities in the Birhan North Shewa Zone, Amhara Region, Ethiopia. The field site was established in June 2018. The Birhan field site is a community-based continuous follow-up study of health and demographic conditions that provides up-to-date information on the catchment population and establishes a population frame to nest studies. We selected all catchment health facilities for this study, including five health centres, two primary hospitals (one public and one private) and one referral hospital. These facilities provide essential MNCH services for both the rural majority population and urban population within the field site catchment and non-catchment areas.

The health centres provide antenatal care (ANC), postnatal care (PNC), delivery, abortion, routine immunisation (RI), integrated management of neonatal and childhood illness (IMNCI), and family planning (FP). Each health centre also has a minimum of five service extension health posts, mainly for FP and RI in each kebele (the lowest administration unit in Ethiopia), and each health post sends monthly activity reports to health centres. Two public hospitals (one primary and one referral) and one private general hospital also provide the aforementioned essential MNCH services, except for RI, which is given mainly in health centres and catchment health posts.

Mixed phenomenological qualitative and facility-based cross-sectional study designs were employed. For the quantitative part of the study, a facility-based cross-sectional survey was conducted to assess the impact of COVID-19 on essential MNCH service provision or utilisation and provider-side barriers to service provision and utilisation in the Birhan field site catchment health facilities. We interviewed 91 MNCH HCPs (doctors, nurses, midwives and clinical officers available at the time of data collection) with uniformly structured questionnaires about their perception of client flow and possible barriers for respective sections. Twelve out of 91 HCPs were working in two MNCH departments and were interviewed twice. In addition to this, we extracted retrospective, healthcare utilisation time-series data from each facility using monthly facility reports and medical registers. Retrospective facilities' service statistics were collected over an 18-month period from March 2019 to August 2020 using Computer Assisted Field Editing. We extracted data from the uniformly structured questionnaires, entered it into the Open Data Kit (ODK), and collected and uploaded the data to the ODK aggregate. The monthly facility reports and medical registers data were collected separately. The health centres' monthly reports include services given in the health posts that are extension sites for the health centre, but the facility registers are exclusively for services given in the health centres.

In addition to the cross-sectional study, we implemented a phenomenological qualitative design using in-depth interviews to assess client and provider-side barriers and enablers to service provision/utilisation in the Birhan field site catchment health facilities. We sampled and conducted in-depth interviews until we reached theoretical saturation. For this section of the study, we interviewed 10 facility or department heads, and 9 mothers (delivered at home or facility and had ANC or missed ANC follow-up). An interview guide with open-ended questions was translated from English to Amharic and was used to elicit the qualitative information from informants. We conducted in-person interviews with facility or department heads, women who visited facilities during COVID-19, and women who delivered at the facilities and phone interviews with women who missed an ANC follow-up or delivered at home. With the permission of the respondents, we recorded all interviews and transcribed all records into English for further analysis. To ensure the safety of the data collectors and participants, data collectors wore masks and practised physical distancing during training and data collection from 2 to 20 November 2020.

The extracted data were exported to Stata V.17.0 for analysis and the average MNCH service utilisation was calculated each month to quantify the changes pre-COVID-19 (March–August 2019) and during the COVID-19 (March–August 2020) pandemic. To control for potential seasonal fluctuations in service utilisation, March–August 2019 and March–August 2020 were considered pre-COVID-19 and COVID-19 periods, respectively. Across all health facilities, we had 48 paired months of observations (6 months for each of 8 facilities) for all essential MNCH variables except for RI, which was only administered at the five health centres (and corresponding extension health posts), resulting in 30 paired months. Errors were found in some cases where medical records were misplaced and data for some months were missing or partially filled. To avoid the effect of missing and partially filled values, analogous months' data from the same facility were excluded from the data analysis. Finally, we compared visits for each MNCH service in the pre-COVID-19 and COVID-19 period using a two-tailed independent sample t-test. We repeated the analysis for the initial 2 months (March–April 2020) of the pandemic and the analogous period (March–April 2019) to examine changes in service utilisation at the onset of the COVID-19 pandemic. We used a significance level of $\alpha=0.05$ for all statistical tests.

In addition to the quantitative metrics listed above, English language transcript data were entered in Dedoose software for qualitative data analysis. After familiarisation with the data, the content of the data was coded line by line for thematic analysis following a framework theory approach to describe and interpret health providers' and communities' perceptions of barriers and enablers to MNCH service provision. The framework approach involves using some pre-assigned themes to initially categorise data while also adjusting and iterating the coding scheme to accommodate newly emergent themes, subthemes, and categories through inductive interpretation.[9] Coded data were examined for potential relationships and themes were also assessed across relevant participant demographic categories to understand different user perspectives. Findings were described under pre-assigned and newly emerged themes.

## Patient and public involvement

As in-person meetings were restricted by local authorities during protocol development and data collection due to COVID-19 pandemic, we were not permitted to involve clients or the public in study design or reporting and dissemination plans of our research.

## RESULTS

We extracted data from three hospitals and five health centres (includes 34 service expansion health posts in the community). Maternal health facility visits for ANC, PNC, facility delivery, and abortion-related services decreased in the time of COVID-19; however, we do not see a statistically significant change. The FP service utilisation in the health centres and hospitals declined from 105.5 to 66.5 visits per month (p=0.051) after the onset of the pandemic. Repeat FP visits significantly declined (p=0.046) while new FP visits did not change. When combining health facilities with community health post data,[10] the new FP visits declined significantly from 43.2 visits per month to 28.5 visits per month (p=0.029) and there was no significant change in repeat FP visits.

Declines in service utilisation were also observed for IMNCI, or sick child visits, defined as a facility visit for a sick child under 5 years old. The mean number of IMNCI visits for sick children under 5 years old declined from 225.0 visits per month in 2019 to 139.8 visits per month in 2020 (p=0.014). This significant decline persists for two age stratifications of IMNCI visits (2 months to under 2 years and 2 years to under 5 years). However, there was no significant change (37.0 to 36.8/month, p=0.982) in child visits for routine immunisations, including bacille Calmette-Guérin (BCG) vaccine, oral polio vaccine, pentavalent vaccine (diphtheria, pertussis and tetanus-hepatitis B-*Haemophilus influenzae* type b), and measles vaccinations (table 1). Similar results were found comparing essential MNCH service over 2 months during COVID-19 and the analogous pre-COVID-19 2-month period (online supplemental table 1).

Ninety-one HCPs working in MNCH services were asked about the client flow during COVID-19. Sixty-seven per cent of the HCPs perceived that client flow decreased and 31% of them believed client flow did not change (online supplemental table 2). Qualitative interviews also supported the observed decrease in client flow, with descriptions of sharper contractions in service utilisation in the first couple of months after the onset of the pandemic but resumed to approximately normal levels over subsequent months.

Table 1 Comparing essential MNCH service utilisation over 6 months between COVID-19 (Mar–Aug 2020) and analogous pre-COVID-19 (Mar–Aug 2019) periods

| Visit type | Mean number of visits/ month over 6 months | | t-statistic | P value | Lower p value* | Upper p value† | Paired observations |
|---|---|---|---|---|---|---|---|
| | 2019 | 2020 | | | | | |
| I. Maternal visit | 376.3 | 321.2 | 1.11 | 0.270 | 0.865 | 0.135 | 48/48 |
| 1. Antenatal care | 208.9 | 181.7 | 0.79 | 0.433 | 0.784 | 0.216 | 40/48 |
| 2. Postnatal care | 26.6 | 19.8 | 1.44 | 0.155 | 0.922 | 0.078* | 30/48 |
| 3. Facility delivery | 90.7 | 84.2 | 0.29 | 0.776 | 0.612 | 0.388 | 41/48 |
| 4. Abortion-related services | 11.8 | 9.8 | 0.56 | 0.578 | 0.711 | 0.289 | 34/48 |
| 5. Overall FP services in health posts, health centres and hospitals | 313.3 | 273.4 | 0.82 | 0.415 | 0.792 | 0.207 | 47/48 |
| 5.1 New FP services | 43.2 | 28.5 | 1.22 | 0.029 | 0.986 | 0.014** | 47/48 |
| 5.2 Repeat FP services | 270.2 | 244.9 | 0.57 | 0.567 | 0.716 | 0.284 | 47/48 |
| 6. FP services in health centres and hospitals | 105.5 | 66.5 | 1.99 | 0.051 | 0.974 | 0.026** | 33/48 |
| 6.1 New FP services | 8.9 | 7.1 | 0.84 | 0.406 | 0.797 | 0.203 | 33/48 |
| 6.2 Repeat FP services | 96.5 | 59.3 | 1.03 | 0.046 | 0.977 | 0.023** | 33/48 |
| 6.3 Unclassified FP services | 17.7 | 1.3 | 1.12 | 0.039 | 0.981 | 0.019** | 26/48 |
| II. Sick child visit (0–5 years) | 225.0 | 139.8 | 1.51 | 0.014 | 0.993 | 0.007*** | 46/48 |
| 1. MNCI visit (<2 months) | 10.8 | 7.7 | 0.82 | 0.412 | 0.794 | 0.206 | 46/48 |
| 2. IMNCI visit (2 months–2 years) | 101.6 | 50.4 | 1.68 | 0.009 | 0.996 | 0.004*** | 46/48 |
| 3. IMNCI visit (2–5 years) | 111.6 | 81.8 | 1.15 | 0.034 | 0.983 | 0.017** | 46/48 |
| III. Routine immunisation visit | 37.0 | 36.8 | 0.02 | 0.982 | 0.509 | 0.491 | 23/30 |
| 1. BCG vaccine | 31.4 | 36.5 | −0.39 | 0.701 | 0.350 | 0.650 | 30/30 |
| 2. Oral polio (0) vaccine | 3.2 | 1.0 | 0.88 | 0.384 | 0.808 | 0.192 | 23/30 |
| 3. Pentavalent (DPT-HepB-Hib) (all types) | 100.4 | 101.5 | −0.05 | 0.958 | 0.479 | 0.521 | 30/30 |
| 4. Measles–1 | 10.3 | 27.5 | −1.99 | 0.051 | 0.026 | 0.974 | 30/30 |
| 5. Vitamin A dose | 8.6 | 5.7 | 0.67 | 0.506 | 0.747 | 0.253 | 14/30 |
| **Other types of visits** | | | | | | | |
| All visits | 2568.9 | 2606.7 | −0.05 | 0.956 | 0.478 | 0.522 | 48/48 |
| Adult outpatient visit | 2121.2 | 2239.7 | −0.17 | 0.868 | 0.434 | 0.566 | 44/48 |

*P<0.10; **p<0.05; ***p<0.01.
*Lower tailed test: mean number of visits $H_1$: $\mu_{2019} < \mu_{2020}$.
†Upper tailed test: mean number of visits $H_1$: $\mu_{2019} > \mu_{2020}$.
BCG, bacille Calmette-Guérin; DPT-HepB-Hib, diphtheria, pertussis and tetanus-hepatitis B-*Haemophilus influenzae* type b; FP, family planning; IMNCI, integrated management of neonatal and childhood illness; MNCH, maternal, newborn, and child health.

## Barriers to service provision and utilisation during COVID-19

Even though essential MNCH service utilisation was largely maintained, clients' fear of acquiring the disease from the facility, travel restrictions, increased transportation costs due to the 'half seat' order by the government, and fear of acquiring the disease on the way to the health facility were the main barriers to service utilisation perceived by HCPs (table 2).

Fear of contracting the disease and lack of access to transportation were the most described barriers (online supplemental table 3). Particularly, during the first few months after the onset of COVID-19 in Ethiopia and the imposition of travel restrictions and other public health measures, heightened community fear of acquiring the disease and high levels of public panic were barriers to facility-based service utilisation. Community members were afraid of contracting the disease in crowded spaces, on public transportation routes to facilities, and at facilities from healthcare workers or other patients, especially as confirmed COVID-19 cases were reported at facilities, causing further delay of care-seeking. One client said: "I have postponed my follow up at that time for fear of

**Table 2** Possible barriers to service utilisation in the time of COVID-19 based on healthcare providers' perception

| Possible barriers to service utilisation in the time of COVID-19 | Count | % |
|---|---|---|
| 1. Fear of acquiring the diseases from the facility | 97 | 94 |
| 2. Travel restrictions | 90 | 87 |
| 3. Increased transportation cost (due to 'half seat' order by the government) | 89 | 86 |
| 4. Fear of acquiring the disease on the way to the health facility | 86 | 83 |
| 5. Lack of transport to the HP/HC site | 72 | 70 |
| 6. Lack of PPE for clients | 67 | 65 |
| 7. Clients' perception of limited implementation of protective measures by healthcare providers | 58 | 56 |
| 8. Healthcare providers' advice to stay at home | 54 | 52 |
| 9. Limited service hours or absence of healthcare workers | 17 | 17 |
| 10. Unavailability of ambulance | 7 | 7 |
| 11. Unavailability of healthcare providers in facilities to provide outreach services | 7 | 7 |
| Total interviews | **103** | |

Twelve healthcare providers were working in different departments and asked twice.
HC, health centre; HP, health post; PPE, personal protective equipment.

acquiring the disease from health professionals and health centers. The same is true for other clients in our area, and some mothers have received their visit in private clinics as we perceived almost all staff were infected."

The economic hardship and half-seat transportation restrictions during COVID-19 prevented some clients from being able to pay for transportation. Clients described lacking money to purchase personal protective equipment (PPE) and one HCP noted that the closed market movement affected people's incomes, as reflected by patients delaying treatment until conditions were more severe or defaulting on treatment.

Lastly, multiple clients described that they might be forcibly quarantined or presumed COVID-19 positive if they were to visit facilities, and this fear also deterred facility visits. HCPs described many challenges related to the underpreparedness of the health system to manage suspected cases of COVID-19. Often these challenges manifested through physical infrastructure constraints and a shortage of guidelines for managing quarantine and isolation centres for suspected COVID-19 cases.

### Enablers of service provision and utilisation during COVID-19
In terms of knowledge of COVID-19, all women had heard about the disease, but a few were in doubt about the existence of COVID-19 in the area, which might be

an enabling factor for facility-based service utilisation. A client respondent said: "I do not believe it exists, especially in our area. It might be real/exist in other areas/countries. Healthcare providers just suspect and take everyone into an isolation/quarantine center, even though they are healthy and free of any signs and symptoms…". Some described that COVID-19 could not affect them because God and/or Mary will protect them, citing the importance of prayer as a protective measure.

Facility adaptations, including training for HCPs, hand washing facilities, physical distancing and awareness creation, and health education given by local authorities, increased client awareness of COVID-19 prevention and facility-based service utilisation amid the pandemic over time (online supplemental table 4).

### Understanding service utilisation trends during COVID-19
The barriers and enablers highlighted in the interviews interact with each other dynamically, as depicted below. Some barriers were more substantial than others, particularly fear of the disease, transportation access and economic-related barriers, while certain pull factors encouraged facility visits, particularly over time as fear subsided, community awareness measures were undertaken, and facilities implemented adaptations to manage both COVID-19 and routine services (figure 1).

### DISCUSSION
We examined the impact of the COVID-19 pandemic on essential MNCH service utilisation by analysing data from health facility records and HCPs' and patients' perspectives. In the context of already poor health outcomes, significant reductions in service utilisation for maternal and child health may have substantial adverse impacts. Essential MNCH services such as FP initiation and sick visits for under 5 years old significantly declined during the COVID-19 pandemic. For maternal health, FP, ANC, PNC, facility delivery, and abortion service utilisation decreased, but the change was not significant during the initial 6 months of the pandemic likely because of a small sample size.

A modelling study of essential maternal and child health interventions across 118 low/middle-income countries over a 6-month period estimated a reduction of services by 9.8%–18.5% and 39.3%–51.9% in the least and most severe scenarios, respectively,[11] due to the COVID-19 pandemic; and in China, health service utilisation declined significantly after the outbreak and all indicators rebounded beginning in March 2020, but most had not recovered to their pre-COVID-19 levels by June 2020.[12] A systematic review o f 81 studies across 20 countries reported a median 37% reduction in overall healthcare utilisation and median 42% reduction reduction in health facility visits.[13] In African countries, ANC, PNC visits and facility deliveries were reduced.[14–17] In Addis Ababa, COVID-19-confirmed cases and public panic were higher than in other areas of Ethiopia, and

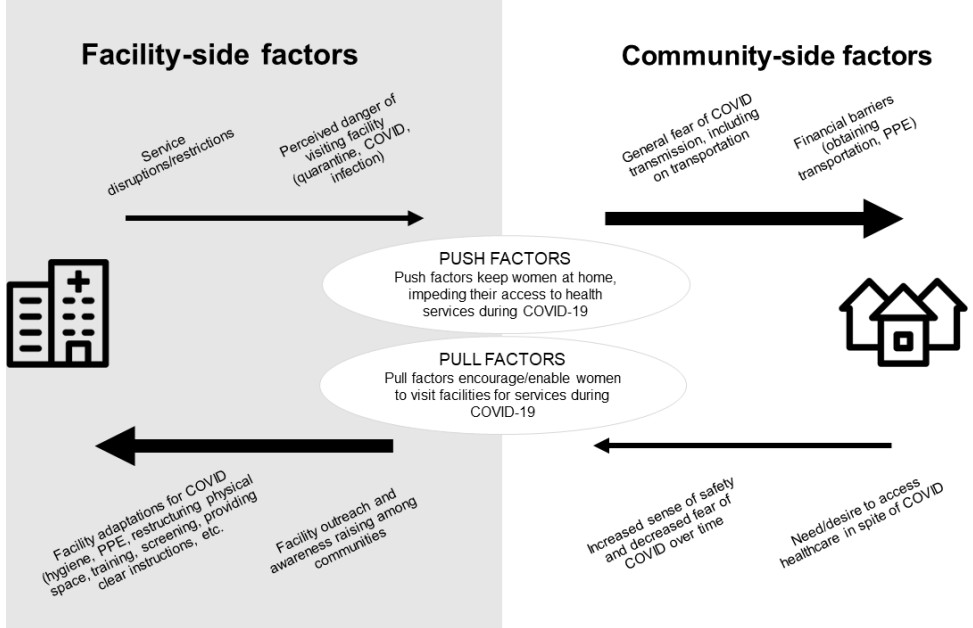

**Figure 1** Enabling (pull) factors and barriers (push factors) for service utilisation highlighted in the qualitative interviews. PPE, personal protective equipment.

women and children's facility-based service utilisation declined.[18] Similarly, a significant reduction of essential MNCH service utilisation was observed from March to June 2020 in Southwest Ethiopia.[19] In contrast, FP, institutional delivery, RI, and ANC did not vary significantly between pre-COVID-19 and during COVID-19 in the Amhara Region, Ethiopia[20] and in Kenya.[21] In our study, maternal facility-based healthcare provision. While there were declines in service utlization, there was a minimun level of service utlization that continued. These findings may be due to the government response to COVID-19, including the absence of a stay-at-home/lockdown policy, relatively low numbers of confirmed cases and death, and facilities open for MNCH services throughout the study period.

Globally, contraception services were shut down or inaccessible[22] and service provision declined.[23 24] In Addis Ababa, FP service utilisation declined for new users and repeat users.[18] In our study, health centre and hospital-based FP service utilisation decreased, but this may have been balanced by services provided by health extension workers at health posts in villages which when combined shows that repeat FP service utilisation was better maintained although still with a modest decline. New FP initiation significantly declined. These results suggest that utilisation of FP services is less likely if visits are new or in more crowded service delivery locations that are farther away requiring extensive travel.

For child health, the number of IMNCI visits, or sick child visits, significantly declined by 38%. Similarly, child health services declined by 33% in three sub-Saharan African countries including Ethiopia due to COVID-19.[24] It is possible that the decrease in child sick visits was related to COVID-19 prevention and control

activities. The leading causes of under 5-year-old children morbidity in Ethiopia,[25] including acute respiratory illness, fever and diarrhoea, may have decreased due to school closures (older siblings less exposed), limited interactions with peers in the community, spending more time indoors, mask-wearing at community gatherings, hand washing, physical distancing, and other PPE and practices. Despite a marked reduction in supply chain distribution of vaccines in Ethiopia during COVID-19,[26] we found that RI remained stable during the initial 6 months of the pandemic, which was different from the findings in Colombia, India, and Brazil where RI declined during the pandemic.[27–29] The MOH of Ethiopia prioritised RI, especially measles, during COVID-19. Existing health extension workers stationed at health posts for community-based services, offering hygiene and sanitation services, FP, and RI, may have sustained accessibility to these services during COVID-19 as they have close relationships with clients and health posts and are often not crowded.

At the time of data collection, early in the pandemic, respondents mainly described not feeling many tangible impacts of COVID-19 on their daily lives, so they conducted daily living activities as usual. This easing fear of COVID-19 may have enabled women to feel that they could safely attend services, but it also has important implications as the pandemic continues, particularly as cases in Ethiopia have risen substantially. Awareness and education campaigns are needed to encourage behaviour change. Moreover, communities' belief that God may protect them from infection indicates the important role of engaging religious leaders as champions in behaviour change campaigns. An additional key recommendation is systematically addressing misinformation and doubt

to increase population compliance with preventive measures, particularly as Ethiopia faces a rising caseload, increasing prevalence of variants, and a stalled vaccine rollout that may take months or years to reach substantial population coverage. Less than 8.4% of the population has received at least one dose of the COVID-19 vaccine as of 19 February 2022.[30]

Barriers to maternal facility visits included women not wanting to bother anyone, lack of support from healthcare workers, the influence of the media,[31] lockdowns, fear of contracting the disease,[32] shift of focus towards the pandemic, resource constraints[14] and non-conducive working environments for HCPs.[33] In addition, women experienced fears of contracting the disease, economic hardship, and lack of access to transportation. Particularly during the first few months after the onset of COVID-19 in Ethiopia, there was an imposition of travel restrictions and other public health measures, like a state of emergency contributing to the high levels of public panic. Facilities restricted some MNCH services at the beginning of the pandemic. Multiple clients described fearing that they might be forcibly quarantined or presumed COVID-19 positive if they were to visit facilities; this fear deterred facility visits. While we found that sick child visits and new FP services were most affected by the pandemic, despite the presence of those barriers, the declines among other essential services were not as significant. Health system resilience and adaptations to maintain provision of services were demonstrated through prioritisation of key interventions, such as immunisation, and reliance on community sources of service provision, such as health posts for FP, to maintain the health system.

## Strengths of the study

We present primary data on service utilisation during the early months of the pandemic in an area of Ethiopia, one of its agrarian regions, which is generalisable to 80% of the country's rural population.[34] We leveraged an existing research network, the HaSET MNCH Research Program (www.hasetmch.org) and our existing Birhan field site.[35] The mixed-methods approach integrated quantitative service utilisation coverage data with sociocultural, contextual and exploratory qualitative interviews to understand trends in service utilisation. The study highlights success stories in community-based care and government leadership for key services like routine immunisation that may benefit other settings.

## Limitations of the study

Our study focused on service utilisation and may not have been powered to detect significant differences. We do not have detailed data on service provision. Recall bias was a potential limitation since qualitative data were collected 3 months after the study period (March–August 2020).

## CONCLUSION

The utilisation of essential MNCH services is crucial to achieving favourable health outcomes. In developing countries like Ethiopia, health systems are often too fragile to withstand the direct increase in the volume of patients and the indirect health consequences of a pandemic. Our study presents early findings on a decline in the utilisation of MNCH services especially in new FP services and sick child visits. Further study is needed to assess the effect of the pandemic on morbidity and mortality among women and children. To sustain health service utilisation during challenging times such as the pandemic, resources are required by government leaders, policymakers and clinicians to improve the resilience of their health system to monitor service utilisation while at the same time engaging with providers and clients to understand and address their evolving concerns about MNCH service uptake.

**Author affiliations**
¹HaSET MNCH Research Program, St Paul's Hospital Millennium Medical College, Addis Ababa, Ethiopia
²Department of Obstetrics and Gynecology, St Paul's Hospital Millennium Medical College, Addis Ababa, Ethiopia
³Department of Pediatrics and Child Health, St Paul's Hospital Millennium Medical College, Addis Ababa, Ethiopia
⁴Epidemiology, Harvard University T H Chan School of Public Health, Boston, Massachusetts, USA
⁵Birhan HDSS, St Paul's Hospital Millennium Medical College, Addis Ababa, Ethiopia
⁶Department of Global Health and Population, Harvard University T H Chan School of Public Health, Boston, Massachusetts, USA
⁷Francis I Proctor Foundation, University of San Francisco, San Francisco, California, USA
⁸Ethiopian Public Health Institute, Addis Ababa, Ethiopia
⁹Boston Children's Hospital, Department of Pediatrics, Harvard Medical School, Boston, Massachusetts, USA

**Acknowledgements** We acknowledge the kind help and encouragement we received from the zone, woreda and study facilities. Our deepest gratitude goes to the women and healthcare providers who participated in this study and to all data collectors and supervisors for their dedicated work. We thank Hayat Alkadir for reviewing the manuscript and providing copy edits.

**Contributors** CB, DB, GJC conceived the idea for the study. CB, DB, BMH, KVW, MK, CT, LT, GJC participated in the design, implementation, interpretation of results. KVW, FAG, MK, CT, GJC participated in the data analysis. CB, FAG, MK, CT, GJC contributed to the initial drafts and all authors were involved in the final draft. DB, LT, GJC were responsible for supervision and mentorship. CB and GJC serve as guarantors of the manuscript.

**Funding** This work is supported by the Bill & Melinda Gates Foundation. The research is co-funded through INV-006752 HaSET and INV-003612 HaSET Ethiopia Partnership.

**Competing interests** None declared.

**Patient and public involvement** Patients and/or the public were not involved in the design, or conduct, or reporting, or dissemination plans of this research.

**Patient consent for publication** Not required.

**Ethics approval** This study involved human participants and the protocol was approved by the Institutional Review Board (IRB) of Saint Paul's Hospital Millennium Medical College (SPHMMC) (PM23/104) and the Harvard T H Chan School of Public Health (HSPH) (IRB20-1574). We obtained permission from all individual health facilities and individual verbal consent from interview respondents.

**Provenance and peer review** Not commissioned; externally peer reviewed.

**Data availability statement** Data are available upon reasonable request.

**ORCID iDs**
Chalachew Bekele http://orcid.org/0000-0003-0348-7566
Grace J Chan http://orcid.org/0000-0002-2716-1643

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
