## [Reviewer comments · BMJ Open]

ARTICLE DETAILS

TITLE (PROVISIONAL)	Impact of the COVID-19 Pandemic on Utilization of Facility-Based Essential Maternal and Child Health Services from March to August 2020 Compared to Pre-Pandemic March to August 2019: a Mixed Method Study in North Shewa Zone, Ethiopia
AUTHORS	Bekele, Chalachew; Bekele, Delayehu; Hunegnaw, Bezawit; Van Wickle, Kimiko; Gebremeskel, Fanos; Korte, Michelle; Tedijanto, Christine; Tadesse, Lisanu; Chan, Grace

VERSION 1 – REVIEW

REVIEWER	Le Doare, Kirsty University of London
REVIEW RETURNED	14-Dec-2021

GENERAL COMMENTS	Thank you for your manuscript which adds important data to the pandemic in MNCH services. Overall, the data presented are strong. However, it is difficult to judge the full extent of your results without a control group. Do you have data from before the pandemic that could be used to show that what you are seeing is a true reflection of the pandemic and not associated with other factors. For severity of lockdown, there are several online resources that can be used and it would be useful to add these to your manuscript as context would be really helpful. 1. Introduction - It would be good to describe in more details the lockdown measure and when they came in, for example, if ANC services were initially closed but then reopened, you would expect to see delay in adverse pregnancy outcomes because of missed visits. 2. Methods - I note in your methods that you have a control group but this is not clear in the abstract so should be added as it will otherwise miss any potential systematic reviews. I think that interrupted time series might be a more robust way to show your data than comparing means as this will take into account any additional confounders and trends over time. You mention this briefly but there are no details. Whilst you did not include participants in the design of your study, you did include them in one of your main objectives, so I would add that. Not every study asks the "why" question. Can you state that your interviews were from both HCW and client perspective. Please include the level of lockdown restrictions in more detail which can be found in many of the online resources such as the Oxford Crisis Index. 3. Results - interrupted time series would be useful to add Is a belief that COVID doesn't exist really an enabler of HCF use? 4. Discussion - you repeat much of your introduction in your discussion, would it be better to focus in your discussion on where you are similar or different to other studies in the region. There are studies from Kenya, Rwanda, Ethiopia and Uganda that you could
--

	compare to. You also include some of your results in your discussion - for example client views on service and these should be moved to your results section and then focus on how these results fit with what is already known. 5. References - several references are missing: Kenya: Shikuku D, Nyaoke I, Gichuru S. Early indirect impact of COVID-19 pandemic on utilization and outcomes of reproductive, maternal, newborn, child and adolescent health services in Kenya. medRxiv 2020:2020.09.09.20191247. Uganda: PMID: 34890417; PMID: 34452941 African: PMID: 34886893; PMID: 32526597
--	---

REVIEWER	Kimani, Rachel The Rockefeller University
REVIEW RETURNED	30-Dec-2021

GENERAL COMMENTS	Thank you for the opportunity to review this manuscript on MNCH utilization in the first six months of the COVID-19 pandemic. This is a valuable paper and contributes to understanding service utilization during health disasters. See comments below. Abstract Introduction- Please give more context apart from the first case date Methods: Indicate which mixed design and a summary of the methodology, including the number of participants in this study Results: Add the statistics for all the mentioned quantitative values, e.g., Antenatal, postnatal, facility delivery New findings: Page 3, Line 43- 45 seems to contradict your results in the abstract. Are MNCH services maintained or decreased? Methods:  1. Are the eight health centers selected the only health facilities in the region, or is there a reason for the sampling?
--

2. Would you please give a clear description of the methodology, including the sequence of quantitative and qualitative arms of the study?
3. Would you please indicate how the cross-sectional survey was sampled? Please define “health care providers”- are these nurses, doctors, community health workers? Indicate numbers for each group
4. What language were the client interviews conducted? Did it require translation?

Results

1. Page 8 Line 31 was the number of healthcare providers interviews 91 or 104? It is unclear what questions were asked of the healthcare provider. Perhaps a table of these results and a list of questions would be helpful in the supplementary.
2. Page 11-line 30 -Supplementary table 3 –, please indicate which words belong to clients and which ones are authors interpretation

Discussion

1. Page12, lines 10-11, the authors suggest that family planning services are more likely to occur when clinics are nearby- have the authors considered alternative explanations of why community-based clinics had stable utilization during the pandemic?
2. Page 12, lines 26-31. In your previous statements, you suggested the reason for maintenance in services was the community-based clinics and services. What results from this study or literature support that facility adaptations maintain services?
3. Page 12, line 27- spelling of “nothing”
4. Page 12, lines 48-49 would you expand to what this sentence refers to. Is this from your study or the literature?
5. In the discussion, please tie in the service utilization results with the cross-sectional surveys and client interviews. For example, if there was no change in utilization, how did push and pull factors noted by HCP and clients interact given the results

Strength and Limitations of the study

1. There are methodological limitations to this study. Given the lack of clarity on sampling, sample size, and analysis, it is unclear how applicable these results are.
2. Would you please address why your results are not generalizable or may be generalizable to your region, country? In the strength of the study, the authors suggest their results are generalizable to 80% of the country population (page 13, line 34). Could you give evidence of this?
3. What steps did you take to address the limitations of this study?

	General Though there is value in the paper, major methodological issues need to be addressed as outlined above. In addition, the manuscript lacks sufficient background/context of how COVID-19 affected the region. For example, how many cases of COVID were in the country, region, the selected hospitals during the study period, and did this affect the results of the study, or how do the results differ from other studies with different results. Furthermore, several tensions in the manuscript should be resolved. 1) It is not always clear whether the authors say there is a reduction in the utilization of services or that the services were maintained. 2) Did government policies enable utilization of services, or were they a barrier 3) Were the clients afraid to visit health facilities during the early stages of the pandemic, or did they did not feel any impact of COVID in their lives?
--	---

VERSION 1 – AUTHOR RESPONSE

II. Reviewer’s comments

Reviewer1: Prof. Kirsty Le Doare, University of London

Comments to the Author:

Thank you for your manuscript which adds important data to the pandemic in MNCH services. Overall, the data presented are strong. However, it is difficult to judge the full extent of your results without a control group. Do you have data from before the pandemic that could be used to show that what you are seeing is a true reflection of the pandemic and not associated with other factors.

Authors’ response: *The quantitative part of the study has a control group. We took 18 months retrospective data March 2018 to Aug 2020 and to avoid the seasonal effect of client flow, we compared the initial six months of the pandemic (Mar -Aug 2020) with previous year (Mar -Aug 2019). This is described in the method section as follows:*

“ Retrospective facilities service statistics were collected over an 18-month period from March 2019 to August 2020 using Computer Assisted Field Editing (CAFE). Data was abstracted by uniformly structured questionnaires and entered to Open Data kit (ODK) collect and uploaded to ODK aggregate..... the average MNCH services uptake was calculated each month to quantify the changes pre – COVID-19 (March to August 2019) and during the COVID-19 (March to August 2020) pandemic. For the purposes of analysis, March to August 2019 and March to August 2020 were considered as pre-COVID-19 and COVID-19 periods, respectively.....” Page 4 and line number: 11 - 35

For severity of lockdown, there are several online resources that can be used and it would be useful to add these to your manuscript as context would be really helpful.

Authors’ response: *In general lockdown or stay-at-home policy was not in place in Ethiopia where the study took place. Please see additional details below.*

1. Introduction

It would be good to describe in more details the lockdown measure and when they came in, for example, if ANC services were initially closed but then reopened, you would expect to see delay in adverse pregnancy outcomes because of missed visits.

Authors’ response: *In general lockdown or stay-at-home policy was not in place in Ethiopia where the study took place. Multiple preventive measures were taken during the initial six month of the pandemic, focusing on social distancing, and wearing mask. Some health facilities were assigned as COVID-19 isolation and quarantine centers, and many suspended conducting elective surgeries and select outpatient services.*

..... “half sit” policies (29 March 2020) decreased maximum occupancy on public transit and increased the cost of transportation when traveling within the region while transportation across regions (the second administration unit next to federal government) was paused completely. Additionally, school closures (March

2020), declaration of state of emergency (08 April 2020), so the local contexts were described

in the introduction. **Page 02 and line number: 26 - 36**

2. Methods

I note in your methods that you have a control group but this is not clear in the abstract so should be added as it will otherwise miss any potential systematic reviews.

Authors' response: *Modified as follows.... the trend of service utilization during the first six months of COVID-19 was compared to corresponding time and data points of the preceding year.*

I think that interrupted time series might be a more robust way to show your data than comparing means as this will take into account any additional confounders and trends over time. You mention this briefly but there are no details.

Authors response: *the interrupted data is related to medical records achieving (misplaced and not accessible during data abstraction) (proportion of paired observations are seen in table one) and the section description revised as follows:*

".... We had 48 paired months observations of essential MNCH variables except RI, which was 30 paired

months, since it was given only in five health centers including extension health posts. The facilities medical records were archived misplaced, and we found some months data missing and partially field. To avoid the effect of missing and partially filled values, analogous months data from the same facility were excluded from the data analysis." **Page 04 and line number: 35 - 39**

While the authors agree that interrupted time series would be an interesting and useful study design for this question, we do not have enough information to sufficiently model trends, which may be subject to seasonality, secular trends, and other patterns that would require several years of historical data to model.

Whilst you did not include participants in the design of your study, you did include them in one of your main objectives, so I would add that. Not every study asks the "why" question. Can you state that your interviews were from both HCW and client perspective?

Authors response: Revised the section and added the study participants as follows:

“ Ninety-one MNCH healthcare providers (doctors, nurses, midwives, and clinical officers) available at the time of data collection in eight health facilities were asked with uniformly structured closed ended questionnaires about their perception of client flow and possible barriers for respective sections. Twelve out of 91 healthcare providers were working in two MNCH departments and interviewed twice. In addition to this, healthcare utilization time-series data from each facility was retrospectively collected from medical records and monthly facility reports.

..... Purposive sampling was implemented, and in-depth interviews were conducted until theoretical saturation was reached. Ten facility or department heads, and nine women (delivered at home/facility, had ANC, or missed ANC follow up) were interviewed” Page 04 and line number: 07 - 23

Please include the level of lockdown restrictions in more detail which can be found in many of the online resources such as the Oxford Crisis Index.

Authors' response: *In general lockdown or stay-at-home policy was not in place in Ethiopia where the study took place. Please see additional details above.*

3. Results

Interrupted time series would be useful to add

Authors' response: *Discussed in method section of the reviewer's comment:*

The interrupted data is related to medical records achieving (misplaced and not accessible during data abstraction)

(proportion of paired observations are seen in table one) and the section description revised as follows:

“.... We had 48 paired months observations of essential MNCH variables except RI, which was 30 paired months,

since it was given only in five health centers including extension health posts. The facilities medical records were archived misplaced, and we found some months data missing and partially field. To

avoid the effect of missing and partially filled values, analogous months data from the same facility were excluded from the data analysis.” **Page 04 and line number: 35 - 39**

Is a belief that COVID doesn't exist really an enabler of HCF use?

Authors response: *Having people who do not believe in COVID-19 existence is not wanted behavior for prevention and control activities, but since the community panic or fear of COVID-19 exposure is one of the possible reasons for lower facility visit, the presence of people who didn't believe in COVID-19 existence during the study time shows that COVID-19 was not limiting factor for those people to visit the facility. So, we considered that folks who don't believe in COVID-19 continued visiting facilities*

4. Discussion

You repeat much of your introduction in your discussion, would it be better to focus in your discussion on where you are similar or different to other studies in the region. There are studies from Kenya, Rwanda, Ethiopia, and Uganda that you could compare to. You also include some of your results in your discussion - for example client views on service and these should be moved to your results section and then focus on how these results fit with what is already known.

Authors' response: *Revised using latest references in the region.*

5. References

Several references are missing:

Kenya: Shikuku D, Nyaoke I, Gichuru S. Early indirect impact of COVID-19 pandemic on utilization and outcomes of reproductive, maternal, newborn, child and adolescent health services in Kenya. medRxiv 2020:2020.09.09.20191247.

Uganda: PMID: 34890417; PMID: 34452941

African: PMID: 34886893; PMID: 32526597

Authors' response: We missed those references since some of them were released after we had submitted and we found them (PMID:34452941, 34886893, 34890417, 32526597 and medRxiv 2020:2020.09.09.20191247) and four others relevant for current manuscript revision. These papers have been cited in the Discussion and added to the References.

1. Nega Assefa AS, Dongqing Wang, Michelle L. Korte, Elena C. Hemler, Yasir Y. Abdullahi, Bruno Lankoande, Ourohire Millogo, Angela Chukwu, Firehiwot Workneh, Phyllis Kanki, Till Baernighausen, Yemane Berhane, Wafaie W. Fawzi, and Ayoade Oduola. Reported Barriers to Healthcare Access and Service Disruptions Caused by COVID-19 in Burkina Faso, Ethiopia, and Nigeria: A Telephone Survey. *Am J Trop Med Hyg.* 2021;105(2):323–330. doi:10.4269/ajtmh.20-1619.
2. Paulo Henrique das Neves Martins Pires CM, Ahmed Abdirazak, Jaibo Rassul Mucufu, Martins Abudo Mupueleque DZ, Ronald Siemens and Celso Fernando Belo. Covid-19 pandemic impact on maternal and child health services access in Nampula, Mozambique: mixed methods research. *BMC Health Serv Res* 21, 860 (2021)doi:<https://doi.org/10.1186/s12913-021-06878-3>.
3. Wendemagegn Enbiale SGA, Meaza Seyum, Dereje Bedanie Hundie, Kassawmar Angaw Bogale, Koku Sisay Tamirat, Mulat Birhanu Feleke, Muluken Azage, Dabere Nigatu, and Henry J. C. de Vries. Effect of the COVID-19 Pandemic Preparation and Response on Essential Health Services in Primary and Tertiary Healthcare Settings of Amhara Region, Ethiopia. *Am J Trop Med Hyg.* 2021;105(5):1240–1246. doi:10.4269/ajtmh.21-0354.
4. Ameyaw EK, Ahinkorah, B.O., Seidu, AA. et al. Impact of COVID-19 on maternal healthcare in Africa and the way forward. 201;223 (2021). doi: <https://doi.org/10.1186/s13690-021-00746-6>.
5. Burt JF OJ, Lubyayi L ea. Indirect effects of COVID-19 on maternal, neonatal, child, sexual and reproductive health services in Kampala, Uganda. *BMJ Global Health.* 2021;6: e006102. doi:10.1136/bmjgh-2021-006102.
6. Eunice Pallangyo MGN, Rose Maina, Valerie Fleming,. The impact of covid-19 on midwives' practice in Kenya, Uganda and Tanzania: A reflective account, *Midwifery*, 2020;89 (0266-6138) doi:<https://doi.org/10.1016/j.midw.2020.102775>
7. Kayiga H GD, Amuge PM, Ssemata AS, Nanzira RS, Nakimuli A. Lived experiences of frontline healthcare providers offering maternal and newborn services amidst the novel coronavirus disease 19 pandemic in Uganda: A qualitative study. *PLoS One*; 10 Dec, 2021.
8. Shikuku1 D, IN, SG, et al. Early indirect impact of COVID-19 pandemic on utilization and outcomes of reproductive, maternal, newborn, child and adolescent health services in Kenya. *medRxiv*; 2021
9. Ethiopia Demographics 2020 (Population, Age, Sex, Trends) - Worldometer (worldometers.info).

Reviewer 2: Dr. Rachel Kimani, The Rockefeller University

Thank you for the opportunity to review this manuscript on MNCH utilization in the first six months of the COVID-19 pandemic. This is a valuable paper and contributes to understanding service utilization during health disasters. See comments below.

1. Abstract

- ✓ Introduction- Please give more context apart from the first case date
- ✓ Methods: Indicate which mixed design and a summary of the methodology, including the number of participants in this study
- ✓ Results: Add the statistics for all the mentioned quantitative values, e.g., Antenatal, postnatal, facility delivery

Authors' response: *Described the pandemic and study design in more detail and added statistics for ANC, PNC, over all FP, RI, and facility delivery in the abstract.*

- ❖ **Introduction:** *..... various measures were taken since then to prevent the transmission of the virus. As a result of the ongoing preventive measures and community fear of exposure, we anticipated that utilization of maternal, newborn and child health (MNCH) services at health facilities would decrease and aimed to assess the MNCH services utilization during the first six months of the COVID-19 pandemic. **Page 01 and line number: 2 - 7***
 - ❖ **Method:** *The study was conducted in all BIRHAN Health and Demographic Surveillance System (HDSS) catchment health facilities in Ethiopia. A mixed facility based cross sectional and a phenomenological qualitative design was used and the trend of service utilization during the first six months of COVID-19 was compared to corresponding time and data points of the preceding year. **Page 01 and line number: 8-14***
 - ❖ **Result:** *..... Antenatal (208.9 to 181.7/month, $P = 0.433$), and postnatal care (26.6 to 19.8/month, $P = 0.155$) visits, facility delivery rates (90.7 to 84.2/months, $P = 0.776$), aver all family planning (313.3 to 273.4/month, $P = 0.415$) and child routine immunization (37.0 to 36.8/month, $P = 0.982$) visits were not significantly affected. **Page 01 and line number: 14-16***
- ✓ **New findings:** Page 3, Line 43- 45 seems to contradict your results in the abstract. Are MNCH services maintained or decreased?

Authors' response: *The MNCH services utilization were maintained over six months compared to the same period in the preceding year, except new family planning initiation and sick under five years old child visits.*

2. Methods:

1. Are the eight health centers selected the only health facilities in the region, or is there a reason for the sampling?

Authors' response: *Revised in the 1st paragraph of the method section as follows:*

"We conducted the study in Birhan Health and Demographic Surveillance System (HDSS) catchment health facilities in North Shewa Zone, Amhara Region, Ethiopia. The HDSS was established by Harvard University (HU) and

Saint Paul's Hospital Millennium Medical College (SPHMMC) in Jun 2018, and it is a community based continuous follow up of health and demographic conditions to give up-to-date information about the catchment population and establishes a population frame to nest studies. There are five health centers, two primary hospitals (one public and one private) and one referral hospital in the area, and all (eight) catchment health facilities were selected for this study. Those facilities provide essential MNCH services for both rural (majority) and urban populations coming from HDSS catchment and non-catchment areas." Page 03 and line number: 26-35

2. Would you please give a clear description of the methodology, including the sequence of quantitative and qualitative arms of the study?

Authors' response: *"Mixed phenomenological qualitative and a facility-based cross-sectional study designs were employed. For the quantitative part of the study, a facility-based cross-sectional survey was conducted to assess the impact of COVID-19 on essential MNCH service provision or utilization and provider-side barriers to service provision and utilization in Birhan HDSS catchment health facilities...."*

In addition to the cross-sectional study, a phenomenological qualitative design utilizing in-depth interviews was implemented to assess client and provider side barriers and enablers to service provision/utilization in Birhan catchment health facilities. Purposive sampling was implemented, and in-depth interviews were conducted until theoretical saturation was reached...." Page 04 and line number: 03-21

3. Would you please indicate how the cross-sectional survey was sampled? Please define "health care providers"- are these nurses, doctors, community health workers? Indicate numbers for each group

Authors' response: *".... Ninety-one MNCH healthcare providers (doctors, nurses, midwives, and clinical officers) available at the time of data collection in eight health facilities were asked with uniformly structured closed ended questionnaires about their perception of client flow and possible barriers for respective sections. Twelve out of 91 healthcare providers were working in two MNCH departments and interviewed twice...." Page 04 and line number: 07-10*

4. What language were the client interviews conducted? Did it require translation?

Authors' response: "... Ten facility or department heads, and nine women (delivered at home/facility, had ANC, or missed ANC follow up) were interviewed. An interview guide with open-ended questions was translated from English to Amharic (**supp interview guide**) and used to elicit the qualitative information from informants and face-to-face interviews were conducted in the facilities with facility/department heads, women who visited facilities during COVID-19 and women who delivered at home. Women who missed an ANC follow up were interviewed by phone..." **Page 04 and line number: 22-28**

3. Results

1. Page 8 Line 31 was the number of healthcare providers interviews 91 or 104? It is unclear what questions were asked of the healthcare provider. Perhaps a table of these results and a list of questions would be helpful in the supplementary.

Authors' response: Discussed above as follows "... Ninety-one MNCH healthcare providers (doctors, nurses, midwives, and clinical officers) available at the time of data collection in eight health facilities were asked with uniformly structured closed ended questionnaires about their perception of client flow and possible barriers for respective sections. Twelve out of 91 healthcare providers were working in two MNCH departments and interviewed twice..." **Page 04 and line number: 07-10** and Attached Supp Interview guide

2. Page 11-line 30 -Supplementary table 3 –, please indicate which words belong to clients and which ones are authors interpretation

Authors' response: Illustrative quotes are words belonging to healthcare providers (HCP) and clients/women (W), and Themes are authors interpretations.

4. Discussion

1. Page12, lines 10-11, the authors suggest that family planning services are more likely to occur when clinics are nearby- have the authors considered alternative explanations of why community-based clinics had stable utilization during the pandemic?

Authors' response: Clients don't need any transportation means to reach those health posts since the health posts availability in the villages. In addition, health extension workers who provide the FP method in the health post are giving hygiene and sanitation services home to home and have close relations with clients and the health posts are not crowded.

2. Page 12, lines 26-31. In your previous statements, you suggested the reason for maintenance in services was the community-based clinics and services. What results from this study or literature support that facility adaptations maintain services?

Authors' response: *The FP services utilization in the health centers and hospitals declined from 105.5 to 66.5 visits per month ($p < 0.05$). Since the FP services are given in health posts, the source of data for FP is aggregated data*

from both health posts and health centers and the overall FP service utilization was maintained (313.3 to 273.4/month,

P = 0.415).

3. Page 12, line 27- spelling of “nothing”

Authors response: *corrected*

4. Page 12, lines 48-49 would you expand to what this sentence refers to. Is this from your study or the literature?

Authors' response: *The section revised as follows:*

*“At the time of data collection, early in the pandemic, respondents largely described not feeling many tangible impacts of COVID-19 on their daily lives if they fail to adhere to preventive measures, so they went about life as usual. This easing fear of COVID-19 may have enabled women to feel that they could safely attend services, but it also has important implications as the pandemic continues, particularly as cases in Ethiopia have risen substantially. Awareness and education campaigns are needed to produce actual behavior change...” **Page 10 and line number: 40 - Page 11 and line 06***

5. In the discussion, please tie in the service utilization results with the cross-sectional surveys and client interviews. For example, if there was no change in utilization, how did push and pull factors noted by HCP and clients interact given the results

Authors' response: *In general, our study showed that maternal facility-based healthcare provision was maintained and there were pulling factors for this, but there were also pushing factors for the facility visit, that is why we noted the pushing and pulling factors in a maintained MNCH service utilization.*

Strength and Limitations of the study

1. There are methodological limitations to this study. Given the lack of clarity on sampling, sample size, and analysis, it is unclear how applicable these results are.

Authors' response: *The sampling and sample size described in detail as shown in the method section and discussed in method section as follows*

*"... Although our samples were eight health facilities available in the HDSS they are likely to be representative because clients and healthcare providers' views were added in addition to six analogous months data abstraction from medical records". **Page 10 and line number: 02-05***

2. Would you please address why your results are not generalizable or may be generalizable to your region, country? In the strength of the study, the authors suggest their results are generalizable to 80% of the country population (page 13, line 34). Could you give evidence of this?

Authors' response: *The study was conducted mainly in rural areas and it is representative for 80% of the rural population of the country.*

3. What steps did you take to address the limitations of this study?

Authors' response:

- *We collected maternal and under five years old children deaths in the facilities, but not in the community and data were not included in the manuscript since they were not representative.*
- *We missed a good opportunity since we don't have detailed overall information about service restrictions in the facilities because the data were collected only from MNCH department staff, charts and clients. But we confirmed that the MNCH services were open during the study period in all facilities.*

General/summary

Though there is value in the paper, major methodological issues need to be addressed as outlined above. In addition, the manuscript lacks sufficient background/context of how COVID 19 affected the region. For example, how many cases of COVID were in the country, region, the

selected hospitals during the study period, and did this affect the results of the study, or how do the results differ from other studies with different results.

Authors' response: Described in the introduction of the manuscript as follows;

“The World Health Organization (WHO) declared coronavirus disease-2019 (COVID-19) a global pandemic on March 11, 2020 and the first case of COVID-19 in Ethiopia was registered on March 13, 2020. Ethiopia was one of the countries with lower COVID-19 prevalence and related death during the study period with 63,367 confirmed cases and 974 deaths in ~119 million population as of 10 Sep 2022.2 Majority of cases were from the city Addis Ababa and only 365 confirmed cases and 8 deaths were register up to 30 Aug 2020 in the zone (the 3rd administration unit of the country) occurred where study was conducted.” **Page 02 and line number: 02-08**

Furthermore, several tensions in the manuscript should be resolved.

1) It is not always clear whether the authors say there is a reduction in the utilization of services or that the services were maintained.

Authors' response: Described above.

2) Did government policies enable utilization of services, or were they a barrier

Authors' response: Relative to other countries, Ethiopian government response for COVID-19 was an enabling for MNCH service utilization, because lockdown/stay at home policy was not in place, facilities were open for MNCH service and due to multiple preventive and control measures the prevalence of confirmed case and death due COVID-19 lower, and we have revised the manuscript based on that.

3) Were the clients afraid to visit health facilities during the early stages of the pandemic, or did they did not feel any impact of COVID in their lives?

Authors' response: Based on the open-ended interviews, women were panicked to visit the facility and some others were not believing in COVID-19 existence and leading their lives as usual.

VERSION 2 – REVIEW

REVIEWER	Le Doare, Kirsty University of London
----------	--

REVIEW RETURNED	14-Mar-2022
GENERAL COMMENTS	my comments have mostly been addressed
REVIEWER	Kimani, Rachel The Rockefeller University
REVIEW RETURNED	01-Apr-2022

GENERAL COMMENTS	Thank you for the opportunity to review this manuscript of relevance to MCH services in Ethiopia. The manuscript is generally well written with additional information from previous versions. However, the manuscript would be enhanced by attention to the following items:  1. Manuscript could benefit from an English editorial review. Multiple spelling and grammar errors. 2. The manuscript remains light on the discussion and the authors claim that the results are generalizable to 80% of the country. Multiple articles have been published in Ethiopia looking at MCH services. How does this compare to the results reported?  -Temesgen, K., Wakgari, N., Debelo, B.T., Tafa, B., Alemu, G., Wondimu, F., Gudisa, T., Gishile, T., Daba, G., Bulto, G.A. and Soboka, B., 2021. Maternal health care services utilization amidst COVID-19 pandemic in West Shoa zone, central Ethiopia. PloS one, 16(3), p.e0249214. -Kassie, A., Wale, A. and Yismaw, W., 2021. Impact of Coronavirus Diseases-2019 (COVID-19) on utilization and outcome of reproductive, maternal, and newborn health services at governmental health facilities in South West Ethiopia, 2020: comparative cross-sectional study. International Journal of Women's Health, 13, p.479. -Gebreegziabher, S.B., Marrye, S.S., Kumssa, T.H., Merga, K.H., Feleke, A.K., Dare, D.J., Hallström, I.K., Yimer, S.A. and Shargie, M.B., 2022. Assessment of maternal and child health care services performance in the context of COVID-19 pandemic in Addis Ababa, Ethiopia: evidence from routine service data. Reproductive Health, 19(1), pp.1-11.
---

VERSION 2 – AUTHOR RESPONSE

Reviewers' comment for Authors'

1. Reviewer 1: Prof. Kirsty Le Doare, University of London

❖ Comments to the Author: my comments have mostly been addressed

2. Reviewer 2: Dr. Rachel Kimani, The Rockefeller University

❖ Comments to the Author: Thank you for the opportunity to review this manuscript of relevance to MCH services in Ethiopia. The manuscript is generally well written with additional information from previous versions. However, the manuscript would be enhanced by attention to the following items:

1. Manuscript could benefit from an English editorial review. Multiple spelling and grammar errors.

➤ Authors' responses: We have worked with native English-speaking colleagues to refine the English in this manuscript.

2. The manuscript remains light on the discussion and the authors claim that the results are generalizable to 80% of the country. Multiple articles have been published in Ethiopia looking at MCH services. How does this compare to the results reported?

➤ Authors' responses: Birhan health and demographic surveillance system was established four years

ago to represent mainly the rural population which is greater than 80% of the country population and this study was done in all catchment health facilities, and that is why we considered the study is generalizable for 80% of the country.

1) Temesgen K, Wakgari N, Debelo BT, Tafa B, Alemu G, Wondimu F, Gudisa T, Gishile T, Daba G, Bulto GA, Soboka B. Maternal health care services utilization amidst COVID-19 pandemic in West Shoa zone, central Ethiopia. PLoS One. 2021 Mar 26;16(3):e0249214. doi: 10.1371/journal.pone.0249214. PMID: 33770120; PMCID: PMC7997037.

➤ Authors' responses: West Shoa zone is the third administration unit in Oromia region, adjacent to our study area north Shewa zone in Amhara region. The study is community based and has no control group to see the effect of COVID-19 on women and children facility-based service utilization. So, we did not consider it as a reference for our study.

2) Kassie A, Wale A, Yismaw W. Impact of Coronavirus Diseases-2019 (COVID-19) on Utilization and Outcome of Reproductive, Maternal, and Newborn Health Services at Governmental Health Facilities in Southwest Ethiopia, 2020: Comparative Cross-Sectional Study. Int J Womens Health. 2021 May 19;13:479-488. doi: 10.2147/IJWH.S309096. PMID: 34040456; PMCID: PMC8141395.

➤ Authors' responses: This study was done in southwest Ethiopia (semi pastoralist population) and essential MNCH services were reduced in the initial phase of the pandemic March – June 2020 and similarly clients and HCPs had similar perception in our study. We have added this relevant paper to the 2nd paragraph of the Discussion. To explore this the quantitative data on the initial two pre-COVID (March to April 2019) months were compared with analogous COVID time months (March to April 2020) and there were no

statistically significant changes in the number of visits for maternal and childhood visits overall, except sick child visit (Supp Table 2).

3) Gebreegziabher SB, Marrye SS, Kumssa TH, Merga KH, Feleke AK, Dare DJ, Hallström IK, Yimer SA, Shargie MB. Assessment of maternal and child health care services performance in the context of COVID-19 pandemic in Addis Ababa, Ethiopia: evidence from routine service data. *Reprod Health*. 2022 Feb 14;19(1):42. doi: 10.1186/s12978-022-01353-6. PMID: 35164776; PMCID: PMC8842853.

➤ Authors' responses: This study was done in the capital of Ethiopia, Addis Ababa and published on 14 Feb 2022. The findings are important to discuss the paper and we have revised the discussion based on it (2nd paragraph of discussion section).